# Differential Gene Expression of Subcutaneous Adipose Tissue among Lean, Obese, and after RYGB (Different Timepoints): Systematic Review and Analysis

**DOI:** 10.3390/nu14224925

**Published:** 2022-11-21

**Authors:** Elena Marisol Cruz-García, María E. Frigolet, Samuel Canizales-Quinteros, Ruth Gutiérrez-Aguilar

**Affiliations:** 1Laboratorio de Investigación en Enfermedades Metabólicas: Obesidad y Diabetes, Hospital Infantil de México “Federico Gómez”, Mexico City 06720, Mexico; 2Unidad de Genόmica de Poblaciones Aplicada a la Salud, Facultad de Química, UNAM/Instituto Nacional de Medicina Genόmica (INMEGEN), Mexico City 14610, Mexico; 3División de Investigación, Facultad de Medicina, Universidad Nacional Autónoma de México (UNAM), Mexico City 04510, Mexico

**Keywords:** obesity, subcutaneous adipose tissue, RYGB, gene expression, gene candidate, transcriptome

## Abstract

The main roles of adipose tissue include triglycerides storage and adipokine secretion, which regulate energy balance and inflammation status. In obesity, adipocyte dysfunction leads to proinflammatory cytokine production and insulin resistance. Bariatric surgery is the most effective treatment for obesity, the gold-standard technique being Roux-en-Y gastric bypass (RYGB). Since metabolic improvements after RYGB are clear, a better understanding of adipose tissue molecular modifications could be derived from this study. Thus, the aim of this systematic review was to find differentially expressed genes in subcutaneous adipose tissue of lean, obese and post-RYGB (distinct timepoints). To address this objective, publications from 2015–2022 reporting gene expression (candidate genes or transcriptomic approach) of subcutaneous adipose tissue from lean and obese individuals before and after RGYB were searched in PubMed, Elsevier, and Springer Link. Excluded publications were reviews, studies analyzing serum, other types of tissues, or bariatric procedures. A risk-of-bias summary was created for each paper using Robvis, to finally include 17 studies. Differentially expressed genes in post-RYGB vs. obese and lean vs. obese were obtained and the intersection among these groups was used for analysis and gene classification by metabolic pathway. Results showed that the lean state as well as the post-RYGB is similar in terms of increased expression of insulin-sensitizing molecules, inducing lipogenesis over lipolysis and downregulating leukocyte activation, cytokine production and other factors that promote inflammation. Thus, massive weight loss and metabolic improvements after RYGB are accompanied by gene expression modifications reverting the “adipocyte dysfunction” phenomenon observed in obesity conditions.

## 1. Introduction

Obesity is defined as abnormal or excessive fat accumulation that presents a risk to health [1]. The obesity rates have increased dramatically in the past decades, with a global prevalence of overweight/obesity of ~30% [2]. It is associated with increased risk of diabetes, hypertension, cardiovascular disease, certain types of cancer, and osteoarthritis, among other abnormalities [3].

The main role of adipose tissue (AT) is the storage of triglycerides when circulating fuels are available, namely lipogenesis. An important molecule involved in adipose lipogenesis is fatty acid synthase (FASN) [4]. On the other hand, lipolysis occurs when energy is limited, and the release of fatty acids from AT triglycerides is fundamental to maintaining peripheral organ function [5]. Additional to triglycerides, cholesterol metabolism and transport in adipocytes, a process mediated by apoprotein E (APOE), is modified during obesity due to an increase in intracellular free cholesterol concentrations causing adipocyte enlargement [6,7]. Thus, AT promotes energy homeostasis partly through the regulation of the lipogenesis and lipolysis processes.

However, it is now well known that AT is not only an energy reservoir, but it also actively participates as an endocrine organ producing peptides or “adipokines” as adiponectin (ADIPOQ), which regulates glucose and fatty acid metabolism, and exerts anti-inflammatory effects in target organs [8,9]. Apart from metabolic and endocrine actions, many other functions of AT, such as angiogenesis, adipogenesis, extracellular matrix dissolution and reformation, steroid metabolism, immune response, and hemostasis have been described [10].

These functions are changed during AT expansion [11], when adipogenesis, defined as preadipocyte differentiation into mature adipocytes, is upregulated [12]. A key factor involved in adipogenesis is peroxisome proliferator activating receptor gamma (PPARγ), which mediates physiological AT growth through an increased cell number or hyperplasia which is accompanied by angiogenesis, regulated adipokine production, a low number of immune cells, and balanced lipolysis and lipogenesis [13,14,15,16].

In contrast to physiological AT expansion, fat accumulation due to chronic energy intake leads to obesity. In these conditions, AT growth is mainly associated with increased cell size or hypertrophy and with lower expression of adipogenic factors as PPARγ [17]. Obese adipose depots are characterized by extracellular matrix remodeling to allow adequate tissue expansion, in addition to increased lipolysis, lower secretion of ADIPOQ, and immune cells infiltration (mainly, accumulation of macrophages) [14,18,19,20]. In these conditions, the innate and adaptive immune system is activated by mobilizing several type of leukocytes (lymphocytes, neutrophils, eosinophils, etc.) and local macrophages [21]. In fact, a “healthy” AT contains around 10–15% of macrophages, but under obesity conditions they increase up to 50% [22]. Polarization and activation of AT proinflammatory macrophages contributes greatly to the chronic low-grade inflammation observed in obesity conditions [23]. The M1 macrophages’ pro-inflammatory responses initiate via the toll-like receptors (TLRs), leading to the signaling pathways mediated by chemokines and cytokines, as tumor necrosis factor (TNF) and interleukins 1 (IL-1) and 6 (IL-6), among other inflammatory mediators. The interleukin signaling provokes the activation of further leukocytes involved in the immune response. One feature of dysfunction is proinflammatory cytokine production from adipocytes or infiltrated macrophages [24]. Signaling of such factors in target cells leads to impaired insulin signaling, diabetes, and metabolic syndrome establishment [25,26].

It has been shown that AT metabolism is greatly modified after massive weight loss induced by bariatric surgery [27]. Bariatric surgery is the most effective treatment for obesity and for sustained weight loss over time [28]. The bariatric gold-standard technique is the Roux en Y gastric bypass (RYGB), and it involves the construction of a gastric pouch and the bypass of the duodenum, reducing gastric volume and nutrient absorption [29]. RYGB yields an estimated weight loss of 73% and remission of diabetes and hypertension in 95 and 81% of cases, respectively [30]. AT biopsies from obese and post-RYGB individuals have been used to evaluate candidate gene expression or transcriptome analysis. These reports have demonstrated gene expression modifications after RYGB in subcutaneous adipose tissue (SAT) with focus on adipocyte volume, lipid metabolism and inflammation [31,32].

Significant weight loss is achieved as early as 4 weeks and 3 months after RYGB; in addition, the loss is maintained at 2–5 years. At these early timepoints (4 weeks and 3 months) diabetes remission indicators such as the HOMA index, oral glucose tolerance tests, and glycated hemoglobin (HbA1c) are significantly lower [33,34]. Another metabolic improvement after RYGB related to adipocyte function is adiponectin concentration, which is attained at one week to 3 months after RYGB [35]. After 2 years, serum adiponectin, adipose lipolysis and fat cell size are still lower than before the surgery, however, at 5 years post-surgery, no further changes in circulating adiponectin were found compared to before surgery, while increased lipolysis was suggested [36].

Some authors have focused on short-term and others on long-term effects of RYGB due to the difficulty and cost of sampling SAT after the surgery. Therefore, this review includes several studies to achieve the comparisons between lean, obese, and different timepoints after surgery. If SAT gene expression in RYGB subjects resembles SAT in lean subjects, then we could confirm that adipocyte functionality is ameliorated after surgery, reverting the “adiposopathy” process observed during obesity.

Thus, the aim of this systematic review was to find differentially expressed genes in the SAT of lean, obese, and post-RYGB individuals at different timepoints.

## 2. Materials and Methods

### 2.1. Protocol and Data

This systematic review and analysis was performed according to the Preferred Reporting Items for Systematic Reviews and Meta-Analyses (PRISMA) guidelines [37]. Complete data as raw values from the microarrays, reported and calculated gene expression, gene expression fold change, among others, can be found in Appendix A.

### 2.2. Information Sources and Search Strategy

Original publications reporting gene expression of SAT before and after RGYB were searched in the electronic bibliographic databases PubMed, Elsevier, and Springer Link using the terms “bariatric surgery” and “AT transcriptome”, “AT RYGB microarrays”, “before and after RYGB microarrays”, “RYGB bariatric surgery RNAseq”, “RYGB SAT transcriptome”, and “bariatric surgery SAT expression or microarrays”. Data from publications between 2015 and 2022 were included. This search was finalized in July 2022. The detailed search approach is described in Figure 1.

### 2.3. Eligibility Criteria

The main criteria for this review were gene expression studies carried out in SAT from lean individuals and obese patients before and after RYGB at different timepoints. The inclusion criteria included studies analyzing mRNA expression of either candidate genes or transcriptomic analysis of SAT from obese patients who underwent RYGB, adults older than 18 years. Studies required gene expression comparison between: (1) obese vs. post- RYGB at different time points, or (2) lean vs. obese. All studies included in our review manifest having obtained the biopsy from the periumbilical regions close to the openings of the trocars during the surgery either by needle aspiration or tissue extraction. When post-bariatric surgery samples were taken the same region (periumbilical) was used for tissue extraction. Thus, the region that was reported to be utilized for gene expression is characterized as a homogenous sample (regarding cell type composition) among individuals. Moreover, we included all the papers meeting the including criteria, regardless of the ethnicity of the patients (Appendix A).

The exclusion criteria were: review papers, studies analyzing serum or another type of AT, other types of bariatric procedures (gastric band, sleeve gastrectomy, RGYB combined with another bariatric surgery, etc.), metabolic or proteomic analysis, DNA methylation pattern or transcriptomic regulation via miRNAs or lnRNAs.

### 2.4. Study Selection

The study selection was composed of four stages. In the screening or identification stage, all bibliographic material was retrieved. Two reviewers screened titles and abstracts independently and duplicates were removed. In the eligibility stage, the full texts of the publications were examined by two authors to assess the eligibility criteria. Any disagreement was solved by consensus. The inclusion stage classified the studies depending on the approach: candidate genes or transcriptomic genes. In the gene identification and analysis stage, we selected the genes and extracted raw data. In addition, analysis of data to compare gene expression between post-RYGB vs. obese and lean vs. obese was performed (Figure 1).

### 2.5. Data Collection and Risk of Bias Assessment

A predesigned extraction sheet was used for relevant information such as: authors, publication year, participants characteristics (number of patients, gender, mean age, and recruitment hospital), mean body mass index (BMI before and after surgery), mean BMI loss, type of bariatric surgery, timepoints post-RYGB, gene expression method used (qPCR or microarray), and change fold gene expression between lean vs. obese and obese vs. post-bariatric surgery (different sample collection time points) (Appendix A). A risk-of-bias summary was created for each paper that was included in this systematic review, analyzing each criteria using the risk of bias visualization tool from Robvis (Appendix A) [38].

### 2.6. Synthesis Methods and Effect Measures

Once the papers meeting the inclusion criteria were selected, and papers matching the exclusion criteria were discarded, the papers were divided in two different approaches: (a) candidate genes (genes with known functions implicated in AT metabolism), or (b) transcriptomic genes (genes obtained from microarrays, but not necessarily with known functions). Raw data for the candidate genes was obtained directly from the figures or tables from each paper. Concerning the transcriptomic genes, microarray data was obtained using the NCBI GEO accession number (GSE66921, GSE53378, GSE84599, GSE72158, and GSE55200) or the raw data contained in the Appendix A. Genes with a *p*-value under *p* ≤ 1 × 10^−8^ were considered for the next stage restricting our analysis to the most significant and differentially expressed genes between groups in each microarray analysis (Appendix A).

Each study presented the gene expression levels relative to either lean, obese or post-bariatric expression and sometimes only raw data was available. Therefore, to be able to compare between different studies, we needed to calculate the fold change expression relative to obese samples values. For this purpose, lean or post-bariatric values were divided by the obese values and 1 was subtracted. The resulting value was named “calculated fold change” = ((lean or post-RYGB)/obese) − 1). The interpretation of these data was: (a) for values higher than 0, the expression of that gene is upregulated in lean and post-bariatric subjects compared to obese subjects; (b) values lower than 0 mean that the expression of that gene is downregulated in lean and post-bariatric subjects compared to obese subjects (Appendix A).

Once we obtained the calculated fold change, a comparison of the gene expression among all the papers of post-RYGB vs. obese and lean vs. obese was performed. Then, only the genes that were differentially expressed between groups, in at least two different papers, were selected. Only the intersected genes between post-bariatric vs. obese and lean vs. obese were selected for the next stage.

A list of genes was created and catalogued in different metabolic pathways. For candidate genes, the gene function and pathway are already known (Figure 2). However, for the transcriptomic genes, the pathway classification was performed using several bioinformatics tools: Gene Ontology (http://geneontology.org/), Reactome (https://reactome.org/PathwayBrowser/, accessed on 14 September 2022), and Topp Gene (https://toppgene.cchmc.org/), (Figure 4).

Finally, graphs were created comparing gene expression fold change among lean, obese and post-RYGB SAT samples at different time points, using Prism V8 (Figures 3 and 5).

## 3. Results

Our initial search identified 375 publications using PubMed, Springerlink, and Elsevier; however, 90 papers were duplicated, and were discarded. The 285 non-duplicated articles were read in detail and 268 were discarded, because inclusion criteria were not met, as shown in Figure 1. Therefore, 17 publications were eligible and included in this systematic review and analysis.

As mentioned above, the main objective of this review is to compare gene expression of SAT before and after RYGB bariatric surgery (different timepoints), as well as with lean samples. Therefore, we selected the publications based on studies comparing post-RYGB vs. obese, as well as lean vs. obese. Then, we divided the publications depending on the methodological approach: (1) candidate genes (qPCR) or (2) transcriptomic genes (microarrays).

Ten studies were selected for the candidate gene approach, based on known gene functions in AT metabolism and those genes were analyzed by qPCR. Five studies compared post-RYGB vs. obese samples, two studies compared post-RYGB and lean vs. obese samples, and only one study compared lean vs. obese samples. Moreover, four studies used both approaches (qPCR and microarray), three studies compared post-RYGB vs. obese, one study compared post-RYGB and lean vs. obese samples, and one study compared or lean vs. obese sample. On the other hand, for the transcriptomic gene approach, one study for each comparison was found: post-RYGB vs. obese, post-RYGB, and lean vs. obese samples.

In the next sections, we will describe these studies and identify the genes that were differentially expressed in SAT between post-RYGB vs. obese and lean vs. obese in at least two studies and from different recruitment countries. A summarized description of each article is presented below, highlighting only the differentially expressed genes for each comparison, i.e., post-RYGB vs. obese or lean vs. obese in at least two studies.

### 3.1. Candidate Gene Approach

In this section, we describe studies that reported candidate genes, because of their known function in AT metabolism. In the papers where microarrays were performed, we took into consideration the candidate genes selection presented by each author.

#### 3.1.1. SAT Candidate Gene Differential Gene Expression between Post-RYGB vs. Obese

Karki et al., 2015 analyzed gene expression by qPCR of SAT samples from 19 obese subjects (≥18 years), with an initial BMI of 42 ± 5 kg/m^2^ and a final BMI of ~10 kg/m^2^ after 8 ± 5 months post-RYGB. It is important to notice that the samples were not taken at the same time, which is a reason why this information should be considered cautiously. The expression levels were measured by qPCR. The selected candidate genes are implicated in lipolysis and triglyceride metabolism and their expression was increased after RYGB and inversely correlated with lipids and glucose markers [39] (Appendix A). From the studied genes, only perilipin 1 (*PLIN1*) was selected for the next stage using our inclusion criteria.

Ortega et al., 2016 analyzed the expression of 17 candidate genes associated with adipocytes differentiation, de novo lipogenesis, fatty-acid transport, glycerol recycling, glucose uptake, insulin sensitivity, adipokines, and inflammation. The samples were obtained from 17 obese individuals with RYGB surgery (initial BMI 45 ± 3 kg/m^2^, age 46 ± 10 years) during the RYGB surgery (90–120 min). At this timepoint, there is no effect on BMI. Therefore, we excluded these data for the following analysis (Appendix A) [40].

Oliveira et al., 2017 analyzed SAT samples from 13 obese patients (~33 ± 9 years) with an initial BMI of ~45 ± 6 kg/m^2^ and reducing about ~10 kg/m^2^ before and after 6 months post-RYGB. They analyzed *UCP2* and *PLIN1* expression by qPCR, reporting that *UCP2* is overexpressed after RYGB. In addition, *UCP2* and *PLIN1* expression influenced the resting metabolic rate in obese individuals, suggesting that they could be used as predictors for percentage of weight loss after RYGB (Appendix A) [41]. However, only *PLIN1* was selected for the next stage of our review.

Katsogiannos et al., 2019 analyzed samples for obese individuals before and after 1 and 6 months post-RYGB. SAT samples were obtained from 13 obese individuals with an initial BMI of 37 ± 4 kg/m^2^, aged 55 ± 9 years. The BMI reduced ~4 kg/m^2^ after 1 month and ~8 kg/m^2^ after 6 months post-surgery, respectively. Genes implicated in glucose metabolism (*SLC2A4* (solute carrier family 2 member4)/*GLUT4, IRS-1*, etc.), adipogenesis, lipid metabolism (ADIPOQ, among others), and cellular proliferation were analyzed by qPCR. They reported that 1 month after RYGB, leptin (*LEP*) was downregulated and *ADIPOQ* (among others) was overexpressed, and even more after 6 months. At 6 months, *PPARG*, *FAS*, *GLUT4*, and *IRS1* were overexpressed (Appendix A) [34]. All these genes were selected for the next stage of our review.

Ferraz-Bannitz et al., 2021 reported SAT differential expression in 13 obese subjects (BMI 42 ± 4 kg/m^2^) aged 38 ± 8 years, before and 3 and 6 months post-surgery, reducing ~6 kg/m^2^ and ~9 kg/m^2^, respectively. The authors analyzed 24 candidate genes involved in inflammation, lipid transport, adipogenesis, aminoacid metabolism, glucose homeostasis and oxidative stress by qPCR. The genes that were differentially expressed, were selected for the next stage of our review were: *ADIPOQ* and *PGC1α* were overexpressed and *PPARG*, *TNFα* (at 6 months), and *IL-6* (at 3 months) were downregulated. Therefore, these authors concluded that bariatric surgery can acutely modulate inflammation and ER-stress (Appendix A) [31].

González-Plaza et al., 2016 analyzed samples from eight obese individuals (52 ± 7 kg/m^2^, ~41 ± 11 years) who, after 2 years, lost ~18 kg/m^2^ (final BMI ~34 ± 9 kg/m^2^). They performed a microarray analysis on more than 62,000 transcripts before and after bariatric surgery. They reported that 172 genes were overexpressed and were implicated in lipids synthesis, Krebs cycle, glucolysis, glucogenolysis, and glucogenesis. Moreover, 731 genes were downregulated and were implicated in the immune system and inflammation. However, the authors validated only 8 candidate genes by qPCR in 20 obese individuals (BMI~56 ± 7 kg/m^2^, ~46 ± 10 years) after two years of RYGB with a loss of ~17 kg/m^2^. These authors concluded that RYGB improves the expression of genes involved in lipid metabolism and in inflammation in SAT, because *IRS-1*, *ACACA*, and *FASN* were overexpressed and *IL-6* and *TNFα* were downregulated (Appendix A) [42]. All these genes were included in the next stage of our review.

González-Plaza et al., 2018 used the data published previously by the same first author in 2016. Then, they subdivided the samples in two groups based on insulin resistance: four low insulin resistant obese individuals (49.5 ± 5.2 kg/m^2^, 39.0 ± 13.1 years, losing ~17.6 kg/m^2^ after 2 years) and four high insulin resistant obese individuals (52.2 ± 6.8 kg/m^2^, 43.2 ± 10.7 years and losing ~15.6 kg/m^2^ after 2 years). For the low resistant group, 227 overexpressed and 619 downregulated genes were reported. For the high resistant group, 153 genes were overexpressed, and 865 genes downregulated. The intersection of these two groups showed that genes implicated in lipids biosynthesis were overexpressed; on the other hand, genes implicated in chemotaxis, immune response, and signaling transduction were downregulated. Specifically, for the low insulin resistant group the overexpressed genes were associated with carbohydrate metabolism, and for the high insulin resistant group the overexpressed genes were implicated in immune response, inflammation, phosphatidil-3-kinase protein, cellular proliferation and differentiation, cytokines interaction, and cancer. The authors concluded that insulin resistance state modifies gene expression after RYGB. Then, only eight genes were validated by qPCR using 20 obese subjects before and after 2 years post-surgery, divided in low insulin resistant (45 ± 13 years, with a BMI loss of ~18 kg/m^2^) and high insulin resistant (47 ± 9 years, with a BMI loss of ~17 kg/m^2^). However, for the validated genes, *IRS-1* was overexpressed and *TNFα* was downregulated post-RYGB in both low and high insulin resistance (Appendix A) [43].

Latorre et al., 2018 analyzed the microarray data previously reported by Ortega et al., 2015 (described in the next section), selecting only 14 candidate genes implicated in inflammation, glucose transport and toll-like receptors, and validated by qPCR. They reported that *IRS1*, *GLUT4*, *ADIPOQ* were overexpressed and *IL6*, *TNFɑ*, *LEP*, and *TLR8* were downregulated 2 years post-RYGB (Appendix A), among other genes [44].

Liu et al., 2016 performed a microarray with 47,231 transcripts from SAT samples of 42 obese individuals (~47 ± 6 kg/m^2^, ~43 ± 11 years), comparing before and 12 months post-RYGB (final BMI 33 ± 5 kg/m^2^, losing ~15 kg/m^2^). After one year, 4236 genes were overexpressed, and 2989 genes were downregulated in post-RYGB. The authors validated the expression of 64 genes involved in the extracellular matrix, collagen accumulation, synthesis and degradation. However, 10 genes were overexpressed and 41 downregulated after 1 year of RYGB (Appendix A). These results suggest that increased collagen degradation and decreased cross-linking are modulated after RYGB, mediating extracellular matrix adaptation after fat mass loss [45]. However, none of these candidate genes were considered for the next stage of our review.

#### 3.1.2. SAT Candidate Gene Differential Gene Expression between Post-RYGB or Lean vs. Obese

Jahansouz et al., 2015 analyzed an acute effect comparing gene expression before and one week after bariatric surgery. They analyzed SAT samples from eight obese individuals (initial BMI ~40 ± 2 kg/m^2^, ~50 ± 3 years) with a loss of ~1 kg/m^2^ after one week. Moreover, they analyzed lean SAT samples (BMI ~24 ± 1 kg/m^2^, 45 ± 6 years). Genes implicated in mitochondrial biogenesis, oxidative stress, and protein carbonylation were analyzed by qPCR (Appendix A). The authors concluded that RYGB increases mitochondrial biogenesis gene expression (like *PGC1α*) in SAT [46], gene that was included for the next stage of our review.

Jürets et al., 2017 reported differential gene expression of SAT samples from lean individuals and obese individuals before and one year after RYGB. The samples were obtained from 20 lean individuals (~25 ± 3 kg/m^2^ de 43 ± 9 years) and from 26 obese individuals (~46 ± 6 kg/m^2^ and ~42 ± 12 years), and after one year of surgery the loss was ~15 kg/m^2^. Expression of 20 candidate genes implicated in inflammation, growth factors, metabolic markers, cellular surface, lipolysis, and apoptosis were analyzed by qPCR (Appendix A). *TNFα* and *CASP3* were overexpressed and *IL6*, *PPARG*, *ADIPOQ*, *CD68*, *PLIN1*, and *GLUT4* were downregulated one year post-RYGB compared to obese individuals [47], and these genes were included in the next stage of our review.

Ortega et al. (2015) performed a microarray with 13,885 transcripts using SAT samples from 16 obese individuals before and after 2 years post-RYGB (BMI ~43 ± 5 kg/m^2^, 48 ± 10 years, losing ~14 kg/m^2^) were analyzed. The results demonstrated the overexpression of 2432 genes implicated in lipids metabolism, energy production, Krebs cycle, oxidative phosphorylation, and mithocondrial disfunction after 2 years of RYGB. Moreover, 2586 genes were downregulated, implicated in lipid metabolism, cell-cell interaction and signaling, and carbohydrate metabolism. They validated 29 genes by qPCR, demonstrating overexpression of *FASN*, *ACLY*, *ACACA*, *ACSL1*, *GLUT4*, *IRS1*, among others, and downregulation of *LYZ*, *TNFα*, *AQP9*, and *IL-6* after 2 years of RYGB. In addition, samples from 26 lean individuals (BMI 24.15 ± 2.3 kg/m^2^, 45 ± 5 years) were also analyzed and compared against obese samples. We realized that most of the genes validated by qPCR presented the same expression in lean as in post-RYGB, except for *ACACA*. However, *TNFα* and *IL-6* were upregulated in controls vs. obese (Appendix A) [48]. Therefore, all the genes described here were included for the next stage of our review.

Petrus et al., 2018 analyzed SAT samples from 16 obese subjects (BMI 41 ± 1 kg/m^2^, 46 ± 2 years) before and after 2 years of RYGB (losing ~16 kg/m^2^), as well as 16 lean individuals (~25 ± 1 kg/m^2^, 48 ± 2 years). Gene expression using a microarray containing more than 28,000 transcripts was performed. The authors focused on associated genes with adipocyte number change in obesity and post-RYGB, obtaining 79 genes. The upregulated genes were growth factors (*TGFB3*, *CXCL2*, *FGF7*, *OGN*, and *PDGFD*) and extracellular matrix organization; on the other hand, the downregulated genes were involved in cell growth and proliferation [49] (Appendix A). However, none of these 5 genes were considered for the next stage of our review.

Kerr et al., 2020 analyzed SAT samples from 50 obese subjects (BMI 43 ± 5 kg/m^2^, 43 ± 9 years) before and after 2 and 5 years post-RYGB (losing ~14 y 11 kg/m^2^, respectively), as well as 28 lean individuals (27 ± 5 kg/m^2^, 44 ± 9 years). At 2 and 5 years post-surgery, only 49 and 38 samples were obtained, and analyzed gene expression using a microarray containing more than 19,000 transcripts. After 2 years of-RYGB, 2420 genes were overexpressed and were involved in processes of protein translation, metabolism and adipocyte differentiation. Moreover, 3576 genes were downregulated and involved in immune and inflammatory responses. In addition, after 5 years of RYGB, 1653 were overexpressed, involved in lipid metabolism, adipocyte differentiation and function, and 3930 genes were downregulated and involved in immune response and cytokines production. For the comparison among lean, obese, and post-bariatric samples, they reported 5583 genes were differentially expressed. However, they were focused in only 60 inflammatory genes. They found that *CD68*, *ITGAM*, *LYZ*, and *TLR8* expression was higher in the obese compared to lean and post-surgery (2 and 5 years) among other genes (Appendix A). In addition, *AQP9* was downregulated, but FASN was upregulated post-RYGB. The authors concluded that, inflammatory gene expression continuously improved after RYGB, despite body weight loss [32]. These genes were considered for the next stage of our study.

#### 3.1.3. SAT Candidate Gene Differential Gene Expression between Lean vs. Obese

After selecting the candidate genes that were differentially expressed between post-RYGB and obese, we wanted to compare their expression between lean and obese. We found one article using qPCR and two with microarray methodology.

Using qPCR, Matulewicz et al., 2017 evaluated genes related to adipogenesis, extracellular matrix remodeling, and inflammation. The samples were obtained from 19 obese individuals (BMI ~33 ± 3 kg/m^2^ aged ~26 ± 5 years), 83 lean individuals (BMI ~22 ± 2 kg/m^2^, aged ~23 ± 2 years). They reported that adipogenic and insulin signaling genes (CEBPB, PPARG, ADIPOQ, IRS-1, IRS-2, GLUT4, among others) were downregulated in obesity. Moreover, *ADIPOQ*, *IRS-1*, *IRS-2*, and *GLUT4* were associated with insulin sensitivity independently of BMI. On the other hand, pro-inflammatory and immune cell marker genes (*CD68*, *ITGAM*, among others) were upregulated in obese compared to lean individuals (Appendix A) [50]. All these genes were considered for the next stage of this review.

Ronquillo et al., 2019 performed a microarray (over 18,000 genes) of SAT samples from eight obese individuals (BMI ~33.03 + −3.15 kg/m^2^, ~41 + −8.59 years) compared to eight lean individuals (BMI ~24 + −1.34 kg/m^2^, ~36 + −9.3 years). Then, validation by qPCR in 37 obese and 35 lean subjects was performed. They showed that 12 genes were overexpressed in the obese, which were implicated in glycolysis, gluconeogenesis, inflammation, insulin, and leptin signaling, lipids and carbohydrate metabolism, and oxidative stress. On the other hand, genes related to endorine function, β-oxidation, lipids synthesis, insulin and leptin signaling were downregulated (Appendix A) [51]. However, for this review we only considered *PPARG*, *ACSL1*, and *IRS2* for the next stage.

In a transcriptomic microarray analysis (more than 10,000 transcripts), Badoud et al., 2017 studied SAT samples from eleven obese subjects (35 ± 1.2 kg/m^2^, 46 ± 1.5 years) and nine lean subjects (BMI 22.1 ± 0.5 kg/m^2^, ~50 ± 3 years). They found 353 differentially expressed genes (130 overexpressed and 91 downregulated) in the obese samples. The overexpressed genes were implicated in cytokine receptor interaction and signaling pathways. On the other hand, the downregulated genes are implicated in lipolysis and AMPc signaling [52]. None of the candidate genes studied in this paper were selected for the next stage of our review. However, the database of the gene expression was used to look for the candidate genes in the next stage.

#### 3.1.4. Analysis of SAT Candidate Gene Expression in Lean, Obese and Post-RYGB (Different Timepoints)

As described above, we analyzed 11 articles where differential gene expression between obese vs. post-RYGB (different timepoints) was reported. We found 21 candidate genes that were common in at least two papers. We classified them upon their function in different metabolic pathways: glucose metabolism (*IRS-1* and *GLUT4*), lipogenesis (*ACSL1*, *ACACA*, *ACLY*, *AQP9*, *ELOVL6*, *FASN*, *SCD1*, and *SREBF1*), lipolysis (*PLIN1*), adipogenesis (*PPARG*), endocrine function (*ADIPOQ* and *LEP*), and inflammation (*CASP3*, *CD68*, *IL-6*, *TLF8*, *TNFα*, *SPP1*, and *LYZ*).

Then, five papers were analyzed for SAT differential gene expression between lean vs. obese samples. We obtained 16 genes that were common in at least two papers between lean and obese. We classified them according to their function in glucose metabolism (*GLUT4*, *IRS-1*, and *IRS-2*), lipogenesis (*ACSL1*, *FASN*), lipolysis (*AQP9*), adipogenesis (*PPARG* and *CEBPB*), endocrine function (*ADIPOQ*), and inflammation (*CCL3*, *CD40*, *CD68*, *IL-6*, *ITGAM*, *TNFα*, and *LYZ*).

We then intersected all of these genes, analyzing the ones that were common between obese vs. post-RYGB and obese vs. lean, finding 11 genes *IRS-1*, *GLUT4*, *ACSL1*, *FASN*, *AQP9*, *PPARG*, *ADIPO*, *CD68*, *IL-6*, *TNFα*, and *LYZ*), shown in Table 1 and Figure 2.

Then, we were interested in comparing and analyzing their expression among different studies and the time after RYGB (Appendix A). We observed that genes involved in glucose and lipid metabolism (*IRS-1*, *GLUT4*, *ACSL1*, and *FASN*) were overexpressed in lean compared to obese subjects. In addition, some of those genes were already overexpressed at 6 months, but all were upregulated at 2 and 5 years after RYGB, consistent in different studies (Figure 3, panels 1–4). For *AQP9*, we observed that in lean and in post-RYGB subjects (2 and 5 years) (Figure 3, panel 5), expression levels were downregulated compared to obese. For *PPARG*, we found that it was overexpressed in lean compared to obese subjects; however, after RYGB there is not a clear trend (Figure 3, panel 6).

For *ADIPOQ*, we observed that the gene expression is barely increased in lean subjects compared to obese ones. Then, in the earlier timepoints (3, 6 months after RYGB) its expression increases from 3–12 fold. However, in 2 and 5 years its expression is reduced to similar levels of lean subjects (Figure 3, panel 7).

For *LYZ*, gene expression was downregulated in lean subjects, increasing for obese ones, but downregulated or reverted after 2 and 5 years post-RYGB (Figure 3, panel 11). Moreover, *CD68* is similar to *LYZ*, but it seems to be downregulated from 12 months after RYGB (Figure 3, panel 8). In addition, two inflammatory gene expressions, *IL-6* and *TNFα*, were inconsistent in lean subjects. However, it is clear that after 3 months post-RYGB their expression is extremely reduced in post-surgery compared to obese subjects (Figure 3, panels 9,10).

These results suggest that RYGB reverts the gene expression of *IRS-1*, *GLUT4*, *ACSL1*, *FASN*, and *ADIPOQ* compared to obese subjects, having an overexpression as compared to lean subjects. On the other hand, *AQP9*, *LYZ*, *CD68*, *IL-6*, and *TNFα* are downregulated in lean, overexpressed in obese, and post-RYGB they are again downregulated, showing that RYGB reverts their expression (Figure 3).

**Figure 3 nutrients-14-04925-f003:**
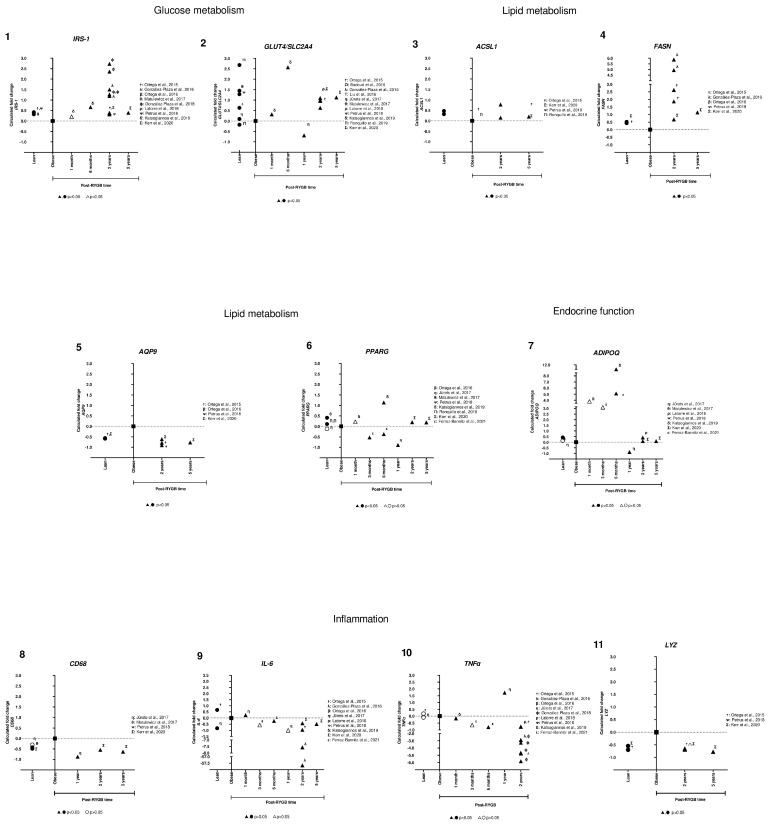
**Subcutaneous adipose tissue candidate gene expression among lean, obese, and post-RYGB (different timepoints).** Comparison of expression levels of each gene among different studies, identified by the Calculated fold change was significant (*p*-value < 0.05, ▲, ●) or not (*p*-value > 0.05, △, ○). (α: [38]; ☨: [47]; Ω: [51]; λ: [41]; Τ: [44]; γ: [40]; η: [46]; θ: [49]; ϕ: [42]; ρ: [43]; ᴪ: [48]; δ: [34]; Π: [50]; Σ: [32]; ε: [31]).

### 3.2. Transcriptomic Gene Approach

In this section, we describe studies that reported SAT gene differential expression obtained by microarrays. As mentioned above, the authors chose the candidate genes based on their known function and implication in AT metabolism. In fact, for some of the microarray papers, the authors focused their discussion on a particular pathway, leading to a limitation of the analysis of the microarray data.

To avoid this limitation, in this section, we obtained the raw data from the microarray of the studies described above. We previously described six papers showing differential gene expression between obese vs. post-RGYB (different timepoints) and two papers comparing obese vs. lean subjects, using a transcriptomic approach (Figure 1). However, from the obese vs. post-RGYB papers, two were discarded from this section (Latorre et al., 2018 and Gonzalez-Plaza et al., 2018), because they used microarray data already published by Ortega et al., 2015 and Gonzalez-Plaza et al., 2016.

For this section, we selected all the genes that had a differential expression with a *p*-value ≤ 1 × 10^−8^.

#### 3.2.1. SAT Differential Gene Expression between Post-RYGB vs. Obese from Microarrays

From the microarray reported by Gonzalez-Plaza et al., 2016, we obtained 2 genes upregulated and 85 downregulated in samples from 2 years post-RYGB compared to obese [42]. Ortega et al., 2015 analyzed samples of obese vs. 2 years post-RYGB. The raw data that we extracted from this microarray showed 21 genes overexpressed and 61 downregulated in post-RYGB [48]. In addition, Liu et al., 2016 studied samples of 1 year post-surgery vs. obesity and we obtained a 369 upregulated and 606 downregulated in post-surgery [45].

#### 3.2.2. SAT Differential Gene Expression between Post-RYGB or Lean vs. Obese from Microarrays

From the microarray reported by Petrus et al., 2018, we obtained 710 genes for 2 years post-RYGB (216 upregulated and 494 downregulated in post-RYGB) [49]. From another database from Kerr et al., 2020, we obtained 1259 genes for 2 years post-RYGB (342 upregulated and 917 downregulated in post-RYGB) and for 5 years post-RYGB, we extracted 1117 genes (213 upregulated and 904 downregulated in post-RYGB) [32].

We then intersected all these genes obtained from microarrays and selected those that were reported in at least 2 papers. We obtained 86 genes differentially expressed in SAT samples between post-RYGB vs. obese.

#### 3.2.3. SAT Differential Gene Expression between Lean vs. Obese from Microarrays

Once we had these 86 genes, we searched them in the raw data from microarrays comparing lean vs. obese published [32,51,52]. We discarded 5 genes, because their expression values were not available [52].

For the 81 genes left, we performed a Gene Ontology analysis (GEO), classifying these genes in different metabolic pathways: lipid and aminoacid metabolism, glycerol transport, acetyl-COA and acyl-COA metabolic process, cell surface, signaling transduction, apoptosis signaling. In addition, most of the genes were classified on inflammation and immune system pathways such as: anti-inflammatory response, innate immune system, inflammation mediated by chemokine and cytokine signaling, interleukins signaling, leukocyte activation and involve in immune response, regulation of leukocyte differentiation, antigen processing cross presentation, neutrophil degranulation, phagocytic cells, and regulation of cytokine production (Figure 4).

After the genes were classified into different pathways, we then analyzed the expression data of lean vs. obese. We discarded 55 genes, because they showed contrary expression levels in lean subjects within different studies.

We ended with 26 genes that are summarized in Table 2. Fold change expression among lean, obese, and post-RYGB (different timepoints) is presented on Appendix A. Then, the expression levels of each gene were plotted, representing lean, obese and post-RYGB different timepoints.

We identified only two genes implicated in lipid metabolism (*FASN* and *APOE*) that were overexpressed in lean compared to obese, and after RYGB the expression was reverted to even higher expression levels (Figure 5, panels 1,2). Interestingly, the rest of the genes showed lower expression in lean, increased in obese, and were downregulated after RYGB involved in: innate immune system (*TLR8*, Figure 5, panel 3), inflammation mediated by chemokine and cytokine signaling pathway (*ALOX5AP*, *PTAFR*, *CCR1*, *LYZ*, Figure 5, panels 4–7), signaling by interleukins (*ITGAX*, *HCK*, *CD4*, Figure 5, panels 8–10), leukocyte activation involved in immune response (*CD83*, *NCKAP1L*, *SELPLG*, Figure 5, panels 11–13), phagocytic cells (*CPVL*, *AOAH*, Figure 5, panels 20, 21), regulation of leukocyte differentiation (*RASSF2*, Figure 5, panel 14), neutrophil degranulation (*BIN2*, *GLIPR1*, *CD300A*, Figure 5, panels 17–19), antigen processing-cross presentation (*CYBB*, *NCF2*, Figure 5, panels 15,16), acetyl-CoA/Acyl-CoA metabolic process (*KYNU*, Figure 5, panel 26), signaling transduction (*ARHGAP30*, *MS4A6A*, *PLEK*, Figure 5, panel 23–25), and apoptosis signaling (BAG3, Figure 5, panel 22).

In general, we observed that expression levels of all these genes were lower in lean compared to obese samples. However, one-year post-RYGB, downregulation of all these genes was similar to lean expression level. Moreover, after 2 years post-RYGB, the expression of these genes reduced even more compared to obese and 1-year post-RYGB (Figure 5).

## 4. Discussion

To our knowledge, this is the first systematic review that compares SAT expression levels between lean and obese (before and after RYGB different timepoints).

In the past decades, bariatric surgery has become an effective intervention to lose weight and revert type 2 diabetes or dyslipidemias. A recent meta-analysis showed that RGYB was one of the most effective bariatric procedures, improving metabolic parameters, such as insulin sensitivity and serum lipids [53]. However, the underlying mechanisms promoting such metabolic changes are unknown, as well as the impact that it exerts over different organs that regulate metabolism, such as AT. In addition, several authors have demonstrated that important adipocyte function features are modified after RYGB, i.e., decreased adipocyte volume [49] (along with decreased lipolysis, immune response and inflammation.

Therefore, the aim of this systematic review was to collect data from SAT gene expression among lean, obese and post-RYGB to start dissecting the pathways and genes influencing obesity establishment and massive weight loss after bariatric surgery.

### 4.1. Candidate Gene Approach

From gene candidate approach, we show that metabolic pathways associated with obesity development and weight loss after bariatric surgery include glucose and lipid metabolism, adipogenesis, endocrine function, and inflammation.

Regarding glucose metabolism, glucose uptake in target cells requires insulin binding to its receptor causing downstream activation of signaling molecules as insulin receptor substrate-1 (IRS-1). After IRS1 activation, GLUT4 can be translocated to the membrane to promote glucose internalization. IRS-1 and GLUT4 expression in AT is typically decreased in obesity and associated with insulin resistance [34,42,43,44,48,54,55,56]. Our findings are consistent with this observation since the expression of *IRS-1* and *GLUT4* is lower in obese individuals. After RYGB, both *IRS-1* and *GLUT4* expression was increased at 6 months, 2, and 5 years. Thus, while insulin signaling molecules expression are decreased in obesity, RYGB increases such expression. After six months, 2, and 5 years insulin sensitivity is augmented and diabetes remission occurs [28,57].

Apart from glucose metabolism, lipid metabolism is also modified after bariatric surgery. ACSL1 (acyl-CoA synthetase long chain family member 1) induces lipid transport and accumulation in adipocytes [58]. These effects are associated with increased insulin sensitivity, since lipid storage in AT prevents lipid accumulation in non-adipose organs. Also, FASN (fatty acid synthase) is a molecular complex that allows fatty acid synthesis from acetyl-CoA [59]. Normally, lipogenesis in AT during obesity is decreased due to chronic energy excess. We found that the expression of both genes was decreased in obese in comparison with lean subjects, and reverted 2 and 5 years after RYGB, suggesting that adipose lipogenesis concurs with metabolic improvements driven by the surgery. On the other hand, lipolysis, which is the hydrolysis of triglycerides to fatty acids and glycerol has been found increased in AT of obese subjects leading to higher circulating levels of free fatty acids. Aquaporins are glycerol and water transporters in AT and liver. Evidence shows that aquaporin 9 (AQP9) could be implicated in adipose lipolysis during obesity conditions [60,61] because lean subjects have decreased *AQP9* expression in contrast to obese patients. After 2 and 5 years of *RYGB*, *AQP9* expression is decreased. Thus, the expression of key molecules involved in lipid metabolism is significantly modified after bariatric surgery, with downregulation of lipolysis and upregulation of lipogenesis, which is similar to metabolism in fat depots of lean subjects.

The transcription factor PPAR*γ* is also implied in lipid metabolism promoting the expression of lipogenic enzymes; however, its main role is to regulate adipogenesis [62]. Since obesity requires adequate energy storage, the recruitment of new adipocytes represents a metabolic advantage. Nevertheless, as chronic energy consumption becomes excessive, the expression of PPAR*γ* and consequent adipogenesis is decreased. These observations are here reflected, since lean subjects have increased PPAR*γ* expression [47,50]. In post-RYGB patients, expression of *PPARγ* after 3–12 months is either down or upregulated, however, 2–5 years after RYGB, this expression appears to be increased. Thus, long-term overexpression of *PPARγ* is consistent with the decreased cell size phenomenon observed after RYGB [27].

One target gene of PPAR*γ* is *ADIPOQ*, which is the most abundant adipokine in circulation. ADIPOQ generates anti-inflammatory effects and is decreased in serum of obese subjects. In fact, hypoadiponectinemia appears to play an important role in obesity-linked insulin resistance [63]. Here we show that *ADIPOQ* expression in AT of lean was lightly increased compared to obese individuals. Also, *ADIPOQ* was 4–8 times overexpressed after 3–6 months of undergoing RYGB. Nevertheless, 2–5 years after RYGB, *ADIPOQ* expression is decreased in comparison with earlier timepoints, but still higher than in obese patients. Finally, short and long-term overexpression of *ADIPOQ* after RYGB could partly explain improved insulin sensitivity and anti-inflammatory processes.

Chronic low-grade inflammation is a feature of obesity, with many related events occurring in AT [64]. The expression of inflammatory markers as CD68, IL-6, TNFα and lysozyme is increased during obesity and associated with insulin resistance [65,66,67,68,69,70]. On the other hand, the lower expression of such factors is related with improved adipocyte function [66,71,72]. We found that the expression of these markers was upregulated in obese individuals and downregulated in post-RYGB patients, especially after 2–5 years. In the short term (before 1 year after RYGB), mixed results regarding up-or downregulation of inflammatory markers was observed. These results suggest that RYGB reduces AT inflammation.

Interestingly, another systematic review analyzed SAT gene expression after lifestyle interventions (diet, exercise, etc.). They observed that *LEP* was downregulated and *ADIPOQ* was upregulated after lifestyle interventions [73]. In our study, LEP was not in the intersected genes, but we observed that *ADIPOQ* was also increased after RYGB. Moreover, for *TNF-α* and *IL-6* expression after lifestyle interventions had contradictory effects [73]. However, our findings showed that for lean subjects the expression was contradictory, but after RYGB both genes were clearly downregulated. This means that weight loss, independent of metabolic modifications driven by the surgery, could regulate gene expression of adipokines, proinflammatory cytokines, among others.

In summary, gene candidate studies show that post-RYGB AT expression reflects a “healthy metabolism” while augmenting insulin-sensitizing molecules, inducing lipogenesis over lipolysis, and downregulating inflammation. Such healthy metabolism is reflected by modification of several genes implicated in these pathways which are reverted few months after RYGB and maintained up to 5 years.

### 4.2. Transcriptomic Gene Approach

As mentioned above, the candidate gene approach is limited to the knowledge of the gene’s function. However, a transcriptomic approach could lead to the discovery of new pathways or genes involved in the AT functionality.

Our results from the transcriptomic approach showed several pathways involved in lipid metabolism (*FASN*, *APOE*), innate immune system (*TLR8*), inflammation mediated by chemokine and cytokine signaling pathway (*LYZ*, *CCR1*, *ALOX5AP*, *PTAFR*), signaling by interleukins (*ITGAX*, *CD4*, *HCK*), leukocyte activation involved in immune response (*CD83*, *NCKAP1L*, *SELPLG*), among others.

We compared the genes from candidate and transcriptomic approach. We identified *CD68* and *AQP9* as candidate genes, but during the transcriptomic analysis, these two genes were part of the 81 differentially expressed genes discarded due to their contradictory expression levels in lean vs. obese subjects. However, *FASN* and *LYZ* were common in both approaches, suggesting an important implication of both genes in AT metabolism in lean and post-RYGB compared to obesity. Moreover, *FASN* is involved in lipogenesis and is downregulated in obese SAT compared to lean and post-RYGB, as explained before. Another gene implicated in lipid metabolism is *APOE*, which controls lipoprotein metabolism. *APOE* is expressed and secreted by adipocytes [74]. *APOE* KO mice present smaller adipocytes and downregulation of lipogenic genes, as well as hypercholesterolemia and atherosclerotic lesions [75,76]. Our results showed higher expression of *APOE* in lean compared to obese and it was upregulated at 1 and 2 years post-RYGB. It has been reported that adipocyte volume is reduced post-RYGB [49]. The exact role of *APOE* in AT and this inverse correlation between expression level and size of the adipocyte has not yet been explained, therefore, it would be important to explore the role of *APOE* in SAT during weight gain and loss

Besides these two genes involved in lipid metabolism, most of the genes that we obtained in our analysis were related to different phases of the innate or adaptative immune response and inflammation. Obesity is characterized by a low-grade chronic inflammation and the mechanism by which the immune response leads to AT hypertrophy remains to be fully elucidated. In addition, little is known of the mechanism by which AT diminishes its size after bariatric surgery. Our transcriptomic results showed common immune response pathways between the expandability and reduction of the AT by comparing lean, obese and post-RYGB are explained below.

First, the initiation of the innate immune response starts with the Toll-like receptors (TLR), which are responsible for the recognition of pathogens or molecular patterns of cellular stress; then, a signaling cascade recruits and activates leukocytes and macrophages [77]. In particular, our results showed that *TLR8* expression was higher in obese SAT compared to lean, as previously reported [78]. *TLR8* has been positively correlated with levels of *IL-6*, *TNFα*, and *CD68* [78]. Interestingly, after bariatric surgery, all these markers were downregulated as well as *TLR8* expression. This suggests that during obesity, *TLR8* must be present to initiate the activation of the immune system, responsible for AT expansion. On the contrary, upon AT size reduction, TLR8 is downregulated at 2 and 5 years post-RYGB, reducing inflammation and AT remodeling.

After the initiation of the immune response, then the secretion of cytokines such as chemokines and interleukins will lead to the mobilization of the leukocytes and macrophages, also known as chemiotaxis, to the site of inflammation. *ALOX5AP* (arachidonate 5-lipoxygenase activating protein) is involved in the synthesis of leukotrienes from arachidonic acid as part of a proinflammatory system. This protein acts together with *ALOX5* to oxygenate and dehydrate fatty acids by sequential reactions, to finally obtain leukotriene B4, which is a chemiotactic molecule. Pharmacological inhibition of ALOX5AP provoked decreased macrophage infiltration, reduced TNF-α, IL6, and free fatty acid concentrations, resulting in a beneficial response [79]. Our results showed that *ALOX5AP* was upregulated in obesity compared to lean and post-RYGB, suggesting that its downregulation, starting at 1-year and maintained up to 5-years post-RYGB, could be responsible for a better AT functionality. Similar expression results were obtained for *PTAFR* (platelet activating factor receptor), which is another chemotactic mediator, involved in angiogenesis and inflammation, but their specific function in AT is still unknown.

Afterwards, the chemokines bind to their specific receptors, for example, CCR1 (chemokine receptor 1). *CCR1* is highly expressed in AT derived stromal cells, participating in the multipotency of these cells and inflammation/migration mediated by chemokine signaling pathway [80]. *CCR1* is highly expressed in obese fat tissue [81], consistent with our results. Moreover, *CCR1* downregulation at 1 and 2 years post-RGYB suggests that macrophage migration would be reduced, as well as inflammation.

A gene that modulates chemokines as well as chemokines receptors is *LYZ*. *LYZ* codes for the lysozyme, generating a proper innate immune system response [82]. *LYZ* was one of the genes that we obtained by the candidate and transcriptomic approach. Our results showed that, *LYZ* was overexpressed in obese AT compared to lean. After 1 year and up to 5 years post-RYGB, its expression was reverted to similar levels of lean AT. *LYZ* function in AT has recently been dissected, demonstrating upregulation in diet-induced obese mice, and being positively correlated with *IL-6* and *TNF-α* (pro-inflammatory cytokines), and *ITGAX* and *CCR2* (macrophage markers). Consistently, when *LYZ* was knockdowned in those animals, increased adipogenesis, decreased inflammation, and improvement of AT functions were observed [72]. This is congruent with our results, where we demonstrated that after RYGB, *LYZ* was downregulated (after 1-year and maintained up to 5-years post-RYGB), as well as *IL-6*, *TNF-α*, and *ITGAX*, indicating that *LYZ* is important for the adequate AT functionality.

*ITGAX* (integrin subunit alpha X) is a member of the integrin family. It is important for chemotaxis and monocyte adhesion, as well as a macrophage marker. During inflammatory responses, *ITGAX* mediates cell-cell interaction. In endothelial cells, *ITGAX* stimulates proliferation, migration and tumor angiogenesis [83]. Our results showed that *ITGAX* expression levels were lower in lean, overexpressed in obese, and downregulated at 2 and 5 years post-RYGB, showing that bariatric surgery reverts its expression. However, its exact function in AT is not yet understood. Another macrophage marker is *HCK* (hemopoietic cell kinase), a member of the Src family of tyrosine kinases. It is expressed in myeloid cells and highly expressed in macrophages. It has been implicated in the signaling of cell surface receptors, regulation of the innate immune response, release of inflammatory molecules, phagocytosis, mobilization of secretory lysosomes, and activation of the NRLP3 (NLR family pyrin domain containing 3) inflammasome [84]. *HCK* is important for macrophage activation and the TNF-α secretion, involved in diabetes progression [85]. Our results showed that *HCK* was downregulated in lean, increased in obese, and reverted to lean levels, after 1 year and maintained up to 5-years post-RYGB. Therefore, this suggests that the macrophage marker, *HCK*, is downregulated in lean and post-RYGB, because of reduced macrophage number.

In addition, adaptive immune response is also regulated during AT expansion and reduction (lean to obese and post-RYGB). One of the genes implicated in this adaptive immune response is *CD4* (CD4 antigen), which encodes a glycoprotein that serves as a surface marker of T-cell and dendritic cells. The CD4+ T-cell surface glycoprotein has been considered another crosstalk molecule between AT and immune system. CD4+ T-cells could secrete pro-inflammatory cytokines, such as interferon (IFN)-γ and IL-17 that will stimulate M1 macrophages, leading to the secretion of IL-6 and TNF-α. On the other hand, CD4+ T-cells could also secrete anti-inflammatory factors (IL-4 or IL-10). Nevertheless, the exact implication of CD4 in AT inflammation is yet to be determined [86]. High *CD4* expression has been associated with higher BMI and obesity-related diseases [87], similar to what we observed in our results between obese vs. lean comparison. Interestingly, this gene was downregulated 1 year after RYGB, and diminished even more at 2 and 5 years post-RYGB. It is plausible that the number of T-cells and dendritic cells decrease after bariatric surgery, as well as their marker *CD4*, but more research is needed to understand its implication in AT functionality.

Another maturation dendritic cell marker and B-cell marker is *CD83* antigen (activated B lymphocytes, immunoglobulin superfamily) [88]. Our results demonstrated that *CD83* is downregulated in lean and post-RYGB (starting at 1-year and maintained up to 5-years) compared to obesity. However, its function in AT is yet to be discovered.

Our results showed several genes implicated in other pathways as: leukocyte activation involved in immune response (*NCKAP1L*, *SELPLG*), regulation of leukocyte differentiation (*RASSF2*), antigen processing-cross presentation (*CYBB, NCF2*), neutrophil degranulation (*BIN2, GLIPR1*, *CD300A*), phagocytic cells (*CPVL*, *AOAH*), apoptosis signaling (*BAG3*), signaling transduction (*ARHGAP30*, *MS4A6A*, *PLEK*), acetyl-CoA/Acyl-CoA metabolic process (*KYNU*). Most of these genes were downregulated in lean, overexpressed in obese, and reverted its expression after 1-year post-RYGB and continuing to reduce its expression at 2 and 5-years after surgery. However, their implication in AT expandability, as well as their specific function in AT is still unknown. Therefore, this opens the horizon of possibilities to explore new pathways and genes and better understand the mechanism how the AT is modulated during the establishment of obesity and during weight loss.

Recently, two studies performed a similar analysis to what we presented in this review. The study performed by Liu et al., 2020, used two database: (1) Hoggard et al., 2012 (obese vs. vertical banded gastroplasty, comparing SAT vs. omental VAT) and (2) Petrus et al., 2018, as previously reported [49,89]. The authors reported similar GO enrichment pathways to our results, such as toll-like receptor signaling pathway, chemokine and cytokine signaling pathway, leukocyte migration and chemotaxis, among others. Moreover, they further analyzed genes methylation and miRNA interaction [90]. The common genes between our study and Liu’s are: *AQP9*, *TLR8* and *HCK*.

The second study performed by Chen et al., 2022, used the database from Hoggard et al., 2012, Ortega et al., 2015, Liu et al., 2016, and Petrus et al., 2018, also obtaining similar results to ours. They showed that the enriched pathways were mainly related to inflammatory response, secretion, defense response, toll-like receptor binding, arachidonic acid binding, neutrophil degranulation and neutrophil activation involved in immune response, among others. In addition, they focused their search on the immunomodulation and immune cell infiltration [91]. The common genes between our analysis and Chen´s were: *AQP9*, *LYZ*, and *APOE*. Even though we found common genes with these two studies, we have to highlight the differences: in our study, we did not consider Hoggard et al., 2022 database, because it did not meet our inclusion criteria. These two papers only compared obese vs. post-bariatric surgery; however, our analysis compared lean vs. obese vs. post-RYGB. Therefore, our results highlight SAT differential expression pathways during gaining or losing weight at different timepoints post-RYGB.

In summary, genes involved in lipid metabolism are highly expressed in SAT from lean, reduced in obese, and upregulated after RYGB (1-year and increased even more after 2-years). In addition, genes involved in several phases of the innate and adaptive immune response are downregulated in lean subjects, elevated in obese and reduced post-RYGB (1-year and further downregulated at 2 and 5-years after surgery).

### 4.3. Strength and Limitation of the Study

The strength of this systematic review was the comparison between two different approaches (candidate and transcriptomic), obtaining a few genes in common. From the transcriptomic approach, we obtained new genes that need be studied to understand their implication in AT functionality. Moreover, the comparisons performed between lean vs. obese and obese vs. post-RYGB highlight the shared pathways and genes of AT plasticity required for weight gain or loss. In addition, showing the gene expression through different timepoints leads to the understanding of processes that happen as acute response, due to body weight loss, or chronic response as the result of sustained weight. Nevertheless, our systematic review has some limitations that must be acknowledge: (1) the candidate approach only identified genes with known functions; (2) each paper reported different sampling timepoints of post-RYGB; therefore, we calculated the fold change from raw databases to compare expression values among studies. Moreover, we could not have the expression of all the genes at all the timepoints reported in several papers; (3) we could not perform meta-analysis, because of the heterogeneity of the samples (individuals number, age, BMI, etc.) and distinct sampling timepoints; (4) several papers did not report the ethnicity of the patients from whom the samples were taken; (5) diet and nutritional data from the patients before sampling is not here described because information was not available, but all the samples were taken under fasting conditions. However, we perform the risk of bias for each paper included in this review and we observed that each one is of high scientific quality, concluding that the results that we present in this manuscript are reliable.

In this systematic review, we summarized the differential gene expression in SAT by comparing lean, obese and post-RYGB. However, we noticed that most of the studies were performed using samples of 1, 2 or 5-years after RYGB. In the future, it would be interesting to perform more studies using samples of previous timepoints, as 1, 3, 6, or 9 months post-RYGB, to understand the acute response on gene expression after this surgery.

## 5. Conclusions

This review summarizes the gene expression patterns in SAT of lean, obese and post-RYGB patients. We conclude obesity leads to adipose tissue dysfunction by promoting impaired adipogenesis and adipokine secretion, increased lipolysis over lipogenesis, insulin resistance, immune response, and inflammation. However, RYGB reverts these effects switching the “unhealthy” features of AT metabolism into a functional version of this organ. These effects are reflected by reverting gene expression post-RYGB, starting 1 year and maintaining such expression up to 5-years after the surgery.

## Figures and Tables

**Figure 1 nutrients-14-04925-f001:**
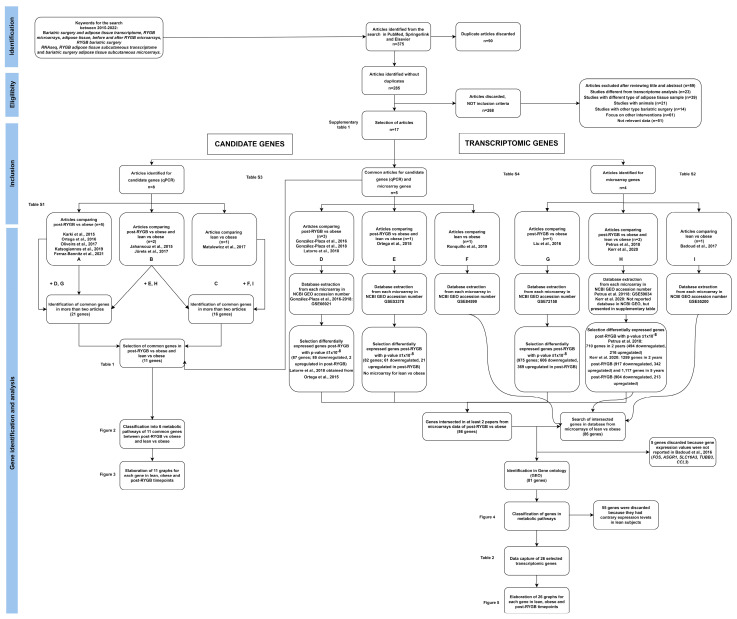
**Systematic review flowchart**. Identification, eligibility, inclusion, gene identification and analysis, using PRISMA (Preferred Reporting Items for Systematic Reviews and Meta-Analyses).

**Figure 2 nutrients-14-04925-f002:**
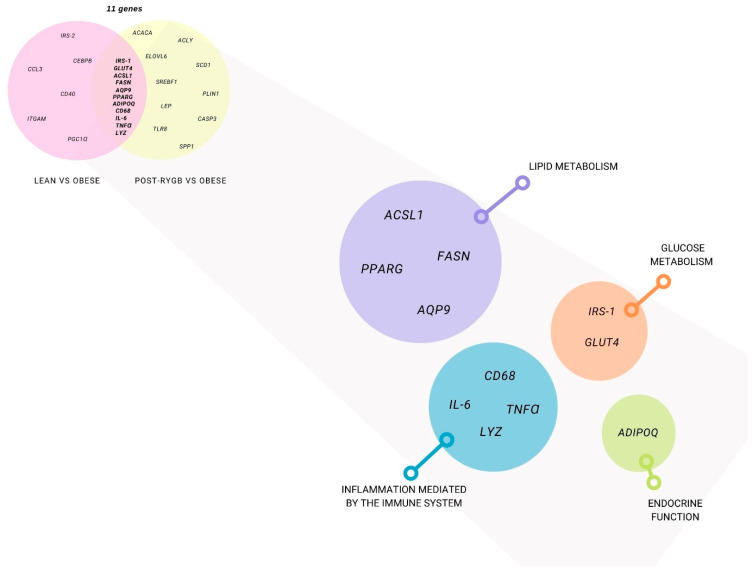
**Subcutaneous adipose tissue differentially expressed genes using a candidate gene approach.** Genes that were studied at least in two papers in post-RYGB vs. obese (yellow) and lean vs. obese (pink) are represented. Gene classification by function: glucose metabolism (orange), lipid metabolism (purple), endocrine function (green), and inflammation (blue).

**Figure 4 nutrients-14-04925-f004:**
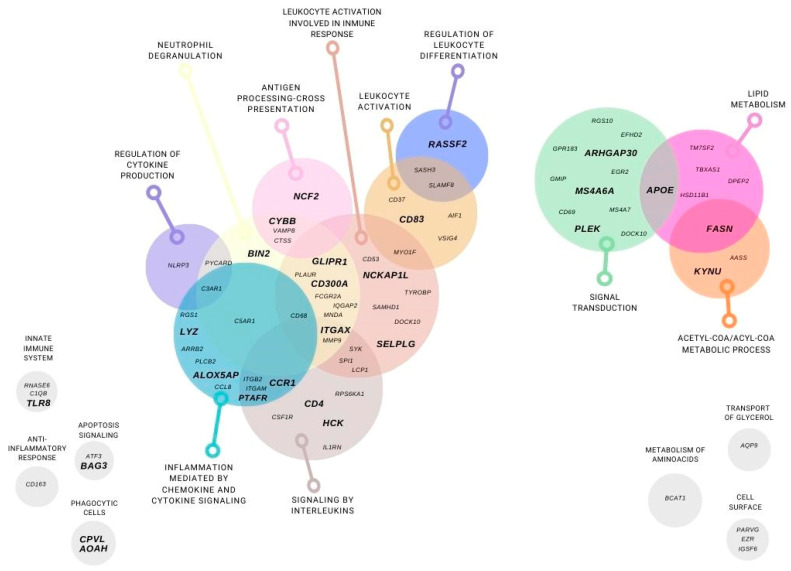
**Subcutaneous adipose tissue differentially expressed genes using a transcriptomic gene approach.** Genes that were differentially expressed in at least two microarrays in post-RYGB vs. obese and were found in lean vs. obese are represented. Gene classification by function: lipid and aminoacid metabolism, glycerol transport, acetyl-COA and acyl-COA metabolic process, cell surface, signaling transduction, apoptosis signaling. In addition, inflammation and immune system pathways such as: anti-inflammatory response, innate immune system, inflammation mediated by chemokine and cytokine signaling, interleukins signaling, leukocyte activation and involve in immune response, regulation of leukocyte differentiation, antigen processing cross presentation, neutrophil degranulation, phagocytic cells, and regulation of cytokine production.

**Figure 5 nutrients-14-04925-f005:**
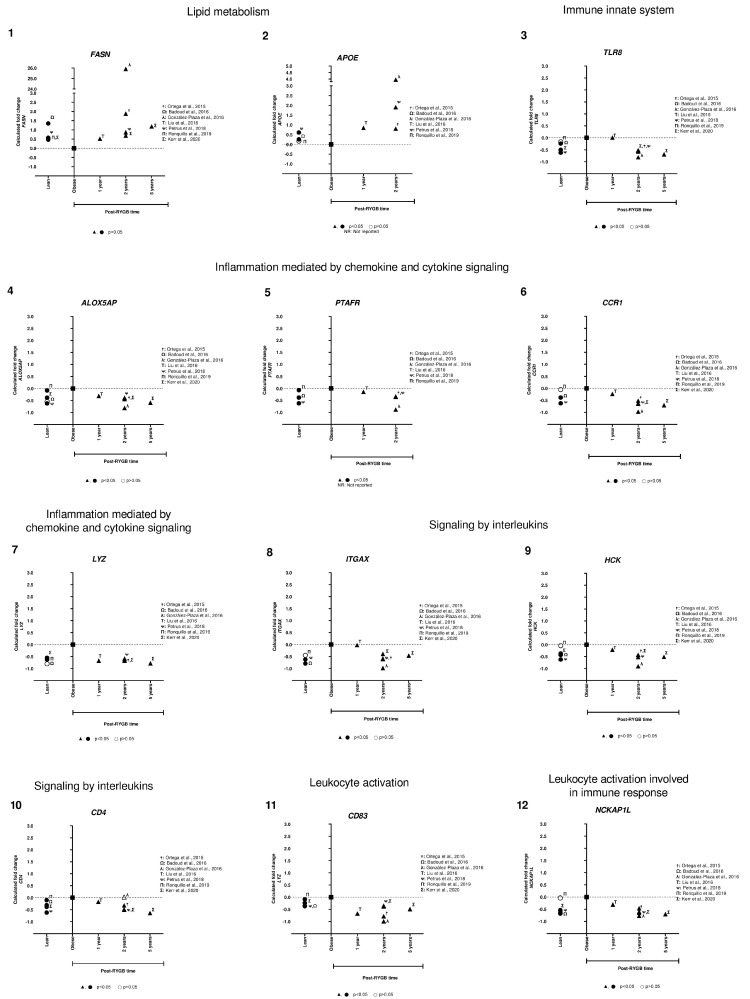
**Subcutaneous adipose tissue transcriptomic gene expression among lean, obese, and post-RYGB (different timepoints).** Comparison of expression levels of each gene among different studies, identified by the author (☨: [47]; Ω: [51]; λ: [41]; Τ: [44]; ᴪ: [48]; Π: [50]; Σ: [32]). Calculated fold change was significant (*p*-value < 0.05, ▲, ●) or not (*p*-value > 0.05, △, ○).

**Table 1 nutrients-14-04925-t001:** Summary of candidate genes differentially expressed in subcutaneous adipose tissue among lean, obese and post-RGYB.

Pathways	Gene Abbreviation	Gene Name
Glucose metabolism	*IRS-1*	Insulin Receptor Substrate 1
	*GLUT4*	Solute Carrier Family 2 Member 4
Lipids metabolism	*ACSL1*	Acyl-CoA Synthetase Long Chain Family Member 1
	*FASN*	Fatty Acid Synthase
	*AQP9*	Aquaporin 9
	*PPARG*	Peroxisome Proliferator Activated Receptor Gamma
Endocrine function	*ADIPOQ*	Adiponectin
Inflammation	*CD68*	Antigen CD68
	*IL-6*	Interleukin 6
	*TNF*α	Tumor Necrosis Factor
	*LYZ*	Lysozyme

**Table 2 nutrients-14-04925-t002:** Summary of transcriptomic genes differentially expressed in subcutaneous adipose tissue among lean, obese and post-RGYB.

Pathways	Gene Abbreviation	Gene Name
Lipids metabolism	*FASN*	Fatty Acid Synthase
	*APOE*	Apolipoprotein E
Innate immune system	*TLR8*	Toll like receptor 8
Inflammation mediated by chemokine and cytokine signaling pathway	*ALOX5AP*	Arachidonate 5-lipoxygenase activating protein
	*PTAFR*	Platelet activating factor receptor
	*CCR1*	Chemokine, cc motif, receptor 1
	*LYZ*	Lysozyme
Signaling by interleukins	*ITGAX*	Integrin Subunit Alpha X
	*HCK*	HCK proto-oncogene, Src family tyrosine kinase
	*CD4*	Antigen CD4
Leukocyte activation	*CD83*	Antigen CD83
Leukocyte activation involved in immune response	*NCKAP1L*	NCK Associated Protein 1 Like
	*SELPLG*	Selectin P ligand
Regulation of leukocyte differentiation	*RASSF2*	Ras association domain family member 2
Antigen processing-cross presentation	*NCF2*	Neutrophil cytosolic factor 2
	*CYBB*	Cytochrome b (−245), beta subunit
Neutrophil degranulation	*BIN2*	Bridging Integrator 2
	*CD300A*	CD300a molecule
	*GLIPR1*	Glioma pathogenesis-related protein 1
Phagocytic cells	*CPVL*	Carboxypeptidase Vitellogenic Like
	*AOAH*	Acyloxyacyl hydrolase
Apoptosis signaling pathway	*BAG3*	BAG cochaperone 3
Signaling transduction	*ARHGAP30*	Rho GTPase activating protein 30
	*MS4A6A*	Membrane-spanning 4-domains, subfamily A, member 6A
	*PLEK*	Pleckstrin
Acetyl-CoA/Acyl-CoA metabolic process	*KYNU*	Kynureninase

## Data Availability

Complete data as raw values from the microarrays, reported and calculated gene expression, gene expression fold change, among others, can be found in Appendix A.

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
