# Peer review of "Differential Gene Expression of Subcutaneous Adipose Tissue among Lean, Obese, and after RYGB (Different Timepoints): Systematic Review and Analysis"

_nutrients, 2022, doi:10.3390/nu14224925_

Round 1

Reviewer 1 Report

The review article “Differential gene expression of subcutaneous adipose tissue among lean, obese, and after RYGB (different timepoints): Systematic review and analysis” describes gene expression alterations related to adipocyte dysfunction. The presented work has been done to find differentially expressed genes in subcutaneous adipose tissue of lean, obese, and post-RYGB (Roux-en-Y gastric bypass). To address this objective the authors analyzed and compared the results of 17 gene expression profiling studies.

The manuscript is well written and contains a valuable summary of current knowledge about gene expression alterations related to obesity conditions. The most valuable part of the review is the meta-analysis of the available gene data sets.

There are points in the presentation and interpretation of the results that need to be revised and expanded:

  1. Please describe in more detail all the statistical procedures used to combine and reanalyze the data. Also please explain how the control of the false discovery rate was applied.
  2. The identified patterns of gene alterations should be interpreted in the context of specific cell types expressing them. It is recommended to discuss this issue and provide limitations of the bulk-tissue approach. 
  3. It is mentioned that the results were analyzed at different time points but there are no conclusions about the time-course of transcriptional alterations activated in response to obesity conditions. It is recommended to discuss the influence of the time factor. 

Author Response

Comments and Suggestions for Authors

The review article “Differential gene expression of subcutaneous adipose tissue among lean, obese, and after RYGB (different timepoints): Systematic review and analysis” describes gene expression alterations related to adipocyte dysfunction. The presented work has been done to find differentially expressed genes in subcutaneous adipose tissue of lean, obese, and post-RYGB (Roux-en-Y gastric bypass). To address this objective the authors analyzed and compared the results of 17 gene expression profiling studies.

The manuscript is well written and contains a valuable summary of current knowledge about gene expression alterations related to obesity conditions. The most valuable part of the review is the meta-analysis of the available gene data sets.

There are points in the presentation and interpretation of the results that need to be revised and expanded:

We thank the reviewer for all your comments. We addressed all the comments made by the reviewer, point by point in the next section:

  1. Please describe in more detail all the statistical procedures used to combine and reanalyze the data. Also please explain how the control of the false discovery rate was applied.

First, we want to clarify that our manuscript is a systematic review, but not a meta-analysis. Therefore, we did not combine the data and not false discovery rate was applied. As a review, we were able to compare the gene expression and summarized it, as well as integrate the expression changes after different timepoints of the bariatric surgery. Each dot on the graphs represents the relative expression levels of each paper included in this review.

We encountered two difficulties to compare expression among the papers. First, each paper reported mRNA expression differently (raw data, expression fold change, or expression fold change – 1). Second, the reference of comparison was different in each paper, being either lean or obese.

To be able to compare the data among different papers, we then extracted all the data and analyzed it in a similar way, and addressed the two difficulties by: 1) considering that all the expression levels will be compared to obese samples (obese or obese before surgery were considered as a relative expression of 1) and 2) the expression of lean or post-bariatric values was divided by the obese values and 1 was subtracted.

Once the data was analyzed, as explained here, it was now comparable among the papers; then, we were able to summarize and create the tables and graphs comparing gene expression of all the papers.

We modified the text as follows (please find the modified text in blue):

  1. 184-195. Each study presented the gene expression levels relative to either lean, obese or post-bariatric expression and sometimes only raw data was available. Therefore, to be able to compare between different studies, we needed to calculate the fold change expression relative to obese samples values. For this purpose, lean or post-bariatric values were divided by the obese values and 1 was subtracted. The resulting value was named “calculated fold change” = ((lean or post-RYGB)/ obese) – 1). The interpretation of these data was: a) for values higher than 0, the expression of that gene is upregulated in lean and post-bariatric subjects compared to obese subjects; b) for values lower than 0 mean that the expression of that gene is downregulated in lean and post-bariatric subjects compared to obese subjects (Table S1, 2, 3).

Once we obtained the calculated fold change, a comparison of the gene expression among all the papers of post-RYGB vs obese and lean vs obese was performed.

  1. The identified patterns of gene alterations should be interpreted in the context of specific cell types expressing them. It is recommended to discuss this issue and provide limitations of the bulk-tissue approach. 

All studies included in our review manifest having obtained the biopsy from the periumbilical regions, close to the openings of the trocars, during the surgery either by needle aspiration or tissue extraction. When post-bariatric surgery samples were taken, the same region (periumbilical) was used for tissue extraction. This means that all biopsies share the features of abdominal subcutaneous tissue, which has been described as homogeneous tissue, regarding its cell type composition, and different from visceral and other adipose depots.

It is known that adipocytes occupy the major tissue area, however, around 60% (subcutaneous) to 80% (visceral) of adipose tissue cells are not mature adipocytes, but preadipocytes, macrophages, stem cells and endothelial cells, as well as neutrophils and lymphocytes. (Tchoukalova,YD, Am J Physiol Regul Integr Comp Physiol, 2004). Specifically, periumbilical adipose tissue (abdominal subcutaneous) is defined as metabolically active with poor collagenic component, few capillaries. (Sbarbati a, 2010, Eur J Hystochem). Thus, the region that was reported to be utilized for gene expression is characterized as a homogenous sample among individuals.

Please find the following text included in the Methods section:

  1. 134-141 All studies included in our review manifest having obtained the biopsy from the periumbilical regions close to the openings of the trocars during the surgery either by needle aspiration or tissue extraction. When post-bariatric surgery samples were taken the same region (periumbilical) was used for tissue extraction. Thus, the region that was reported to be utilized for gene expression is characterized as a homogenous sample (regarding cell type composition) among individuals.

  1. It is mentioned that the results were analyzed at different time points but there are no conclusions about the time-course of transcriptional alterations activated in response to obesity conditions. It is recommended to discuss the influence of the time factor. 

We appreciate the suggestion that the reviewer made, because as he/she highlights, time-course alterations are important to understanding the molecular mechanisms how RYGB leads to a healthier adipose tissue. We are grateful that he/she indicated that for some genes, we were missing this discussion.

We point out the paragraphs where the timepoints were already discussed, as well as the new paragraphs added.

Paragraphs that were already discussed:

L.621-624

L.631-633

L.639

L.648-652

L.657-569

L.666-669

L.706-708

L.748-749

L752-754

L.766-768

L.790-793

New paragraphs added:

L.682-684

Such healthy metabolism is reflected by modification of several genes implicated in these pathways which are reverted few months after RYGB and maintained up to 5-years.

L.726-729 This suggests that during obesity, TLR8 must be present to initiate the activation of the immune system, responsible for AT expansion. On the contrary, upon AT size reduction, TLR8 is downregulated at 2 and 5 years post-RYGB, reducing inflammation and AT remodeling.

L.738-741 Our results showed that ALOX5AP was upregulated in obesity compared to lean and post-RYGB, suggesting that its downregulation, starting at 1-year and maintained up to 5-years post-RYGB, could be responsible for a better AT functionality.

L.759-762 This is congruent with our results, where we demonstrated that after RYGB, LYZ was downregulated (after 1-year and maintained up to 5-years post-RYGB), as well as IL-6, TNF-α, and ITGAX, indicating that LYZ is important for the adequate AT functionality.

L.776-777 Our results showed that HCK was downregulated in lean, increased in obese, and reverted to lean levels, after 1 year and maintained up to 5-years post-RYGB.

L.796-798 Our results demonstrated that CD83 is downregulated in lean and post-RYGB (starting at 1-year and maintained up to 5-years) compared to obesity. However, its function in AT is yet to be discovered.

L.804-806 Most of these genes were downregulated in lean, overexpressed in obese, and reverted its expression after 1-year post-RYGB and continuing to reduce its expression at 2 and 5-years after surgery. 

L.832-836 In summary, genes involved in lipid metabolism are highly expressed in SAT from lean, reduced in obese, and upregulated after RYGB (1-year and increased even more after 2-years). In addition, genes involved in several phases of the innate and adaptive immune response are downregulated in lean subjects, elevated in obese and reduced post-RYGB (1-year and further downregulated at 2 and 5-years after surgery).

In discussion: Section 4.3 Strength and limitation of the study, we also added:

L.850-552 Moreover, we could not have the expression of all the genes at all the timepoints reported in several papers;

  1. 859-863 In this systematic review, we summarized the differential gene expression in SAT by comparing lean, obese and post-RYGB. However, we noticed that most of the studies were performed using samples of 1, 2 or 5-years after RYGB. In the future, it would be interesting to perform more studies using samples of previous timepoints, as 1, 3, 6, or 9 months post-RYGB, to understand the acute response on gene expression after this surgery.

In conclusion: L.871-872 These effects are reflected by reverting gene expression post-RYGB, starting 1 year and maintaining such expression up to 5-years after the surgery.

Reviewer 2 Report

This work is very interesting because it shows at the genetic level the changes that occur after interventions such as RYGB, and how these changes are maintained up to 5 years later.

Regarding this, it would have been interesting to describe if they have found results after 5 years. (line 100 and in discussion too).

In materials and methods, I suggest that you indicate in the "2.4.study selection" section the type of ethnicity of the population as well as some more specific characteristics. In several of the studies you mention, very few patients participate, and these genetic differences may be different for other reasons if they are not well described. If it has been taken into account, it is important to indicate it further. (line 140)

Also in "2.6. synthesis methods" would indicate for the reason that the results with significance values of p>1x10-8 have been chosen. At least, to mentioned better.

In results, also the same as I mentioned above. It would indicate the traits and characteristics of the study subjects better, since it deals with genetics and it is important to specify. (line 197).

It would also be interesting to mention the type of diet and lifestyle that have been described in the studies of these patients. Since this influences gene expression and would also be traits to take into account. (line 660)

Author Response

This work is very interesting because it shows at the genetic level the changes that occur after interventions such as RYGB, and how these changes are maintained up to 5 years later.

Regarding this, it would have been interesting to describe if they have found results after 5 years. (line 100 and in discussion too).

The papers used for this review included samples up to 5 years after bariatric surgery, as it was presented in the results and discussed.

We didn’t find any other paper with the inclusion criteria that reports transcriptomic analysis later than 5 years after the surgery. Two main possibilities could be the cause for not having studies of more than 5 years: 1) no follow-up, 2) bioethical issues. For the follow-up issue, it is well known that after several years after a treatment, in this case a bariatric surgery, a lot of the patients do not return to their checkups. Therefore, less samples are collected as longer the time passes.

On the other hand, certain bioethical issues could be raised by the Ethical Committees for taking a sample after 5 years. Normally, the samples are taken when the patients go back for skin resection or for another intervention due to health issues. For collecting the samples, the patients must be anesthetized, and the sample should be taken on the same area as previous one. For this reason, samples cannot be taken just for research purposes, because it implies a risk for the patient.

In materials and methods, I suggest that you indicate in the "2.4.study selection" section the type of ethnicity of the population as well as some more specific characteristics. In several of the studies you mention, very few patients participate, and these genetic differences may be different for other reasons if they are not well described. If it has been taken into account, it is important to indicate it further. (line 140)

Thank you for your comment. To address this suggestion, we included in Table S1 the place of recruitment of the patients. We could not find the ethnicity in all the papers, as it was not reported. Therefore, we just mentioned the city and country of recruitment.

In our manuscript we added:

In section 2.4:  L.140-141 “Moreover, we included all the papers compiling the including criteria, regardless the ethnicity of the patients (Table S1).”

In section 2.5: L.165 participants characteristics (number of patients, gender, mean age, and recruitment hospital),

In results: Table S1, we added the city and country of recruitment.

In discussion: Section 4.3 Strength and limitation of the study: L.853-854  “4) several papers did not report the ethnicity of the patients from whom the samples were taken;”

Also in "2.6. synthesis methods" would indicate for the reason that the results with significance values of p>1x10-8 have been chosen. At least, to mentioned better.

In genomic and transcriptomic studies, false-positive results could occur, due to multiple comparisons. Normally, to have a significant value, the p-value should be lower than 0.05 (alpha 5%), meaning that 5% of the times, there could be a false-positive result, error type I. When multiple comparisons are performed, the false-positives could increase, causing wrong conclusions of our results. To avoid the false-positive results, and correct error rates while testing a hypothesis in multiple comparison, the Bonferroni test is a simple and strict statistical analysis that could be used. It consists of dividing the alpha by the number of tests performed.

When we analyzed each of the papers with transcriptomic data, we observed that some microarrays analyzed around 20,000 up to 540,000 transcripts. Therefore, the p-value that could be used for analyzing all the papers was 1x10-8, making our analysis strict to the most significant genes in each microarray analysis. By using this approach, we could identify genes that are the most differentially expressed between lean vs obese or post-RYGB vs obese, and classify them in different metabolic pathways.

We added to the manuscript:

L.180-182  Genes with a p-value under p≤1x10-8 were considered for the next stage making our analysis strict to the most significant and differentially expressed genes between groups in each microarray analysis (File S1).

In results, also the same as I mentioned above. It would indicate the traits and characteristics of the study subjects better, since it deals with genetics and it is important to specify. (line 197).

As mentioned above, we added to Table S1 the place of recruitment, because the ethnicity was not reported. Moreover, in section 3 we added:

L.228-230 “In the next sections, we will describe these studies and identify the genes that were differentially expressed in SAT between post-RYGB vs obese and lean vs obese in at least 2 studies and from different recruitment countries.”

It would also be interesting to mention the type of diet and lifestyle that have been described in the studies of these patients. Since this influences gene expression and would also be traits to take into account. (line 660)

The papers included in this review reported gene expression results, however, since lifestyle changes were not among the variables to explore, only few studies manifest the nutritional approach. When they do, they only specify that patients maintained a stable weight one year or three months prior to surgery. Since bariatric surgery and programmed surgeries (biopsy extraction) were always performed under fasting conditions, the subcutaneous adipose tissue biopsy was obtained without the plausible acute gene expression modifications mediated by diet.

We added to our manuscript:

In discussion: Section 4.3 Strength and limitation of the study:  L.854-856  5) diet and nutritional data from the patients before sampling is not here described because information was not available, but all the samples were taken under fasting conditions.
